

# Evaluation of the effect of a communication skills course on medical students' attitude towards patient-centered care: a prospective study

Khalid Gaffer Mohamed[1,2], Amer Almarabheh[1], Asim Mukhideer Almughamsi[3], Hany Atwa[4,5] and Mohamed Hany Shehata[1]

[1] Department of Family and Community Medicine, College of Medicine and Health Sciences, Arabian Gulf University, Manama, Bahrain
[2] Department of Family and Community Medicine College of Medicine, Taibah University, Medina, Saudi Arabia
[3] Department of Surgery, College of Medicine, Taibah University, Medina, Saudi Arabia
[4] Department of Medical Education, College of Medicine and Health Sciences, Arabian Gulf University, Manama, Bahrain
[5] Medical Education Department, Faculty of Medicine, Suez Canal University, Ismailia, Egypt

Corresponding author
Khalid Gaffer Mohamed,
Khalidgm@agu.edu.bh

## ABSTRACT

**Background:** This study aims to examine whether a communication skills course (Early Clinical Exposure–ECE) might support shaping medical students' attitudes towards patient-centered care at Taibah University, Saudi Arabia. Attitudes were measured before and after the course using the Patient Practitioner Orientation Scale (PPOS), which assesses students' orientation towards patient-centeredness.
**Methods:** This prospective observational study collected data at two points: the start of year two and the start of year four from the same cohort. The PPOS, divided into the sharing subscale (physician shares information, power, and control with the patient) and the caring subscale (physician considers the patient's emotions, preferences, and expectations), was used to evaluate students' attitudes. Mean scores were calculated, and independent sample t-tests were employed to assess statistical significance.
**Results:** Of the students targeted, 153 participated at year two (response rate = 93.5%) and 143 at year four (response rate = 89.5%). Significant progress was observed in the sharing subscale, with scores increasing from 3.23 ± 0.53 to 3.40 ± 0.60 ($p = 0.015$). However, the caring subscale exhibited a significant decline, with scores dropping from 4.06 ± 0.53 to 3.76 ± 0.68 ($p < 0.001$). As a result, there was no significant overall change in the total PPOS score ($p = 0.282$). Female students showed more progress in the sharing subscale than males.
**Conclusions:** The communication skills course might have had a positive impact on students' attitudes towards sharing information in decision-making with patients, but notably, the medical students' caring attitudes declined. This indicates a need for further emphasis on patient-centered communication, especially on the caring aspects, in the medical curriculum.

# INTRODUCTION

The concept "People-centered health services" is getting more widely used in recent decades, the World Health Organization (WHO) stated this concept on its strategic reports and policy frameworks (*World Health Organization, 2007*). Moreover, the WHO regarded people-centered care as one of the seven major attributes for health-care quality (*World Health Organization, 2020*). Several rationales were used for this consideration including the higher levels of education people are gaining, the wide availability of information for population and the increasing respect for patient satisfaction.

In clinical practice, the term "Patient-Centered Care" emerged because of the global evolution of attitudes and societal changes that took place in the last century (*Balint, 1969*), surpassing the clinical practice from the one-dimensional biomedical approach to a more holistic approach, taking in account patient's emotions, feelings, concerns, and preferences. This was first expressed by Balint as "patient-oriented medicine" which stands in contrast to "illness-oriented medicine" (*Starfield et al., 1981*). There are several justifications and good evidence supporting the importance of patient-centered methods in medical care (*Balint, 1969*). However, Professor Ian R. McWhinney, the founding father of Family Medicine in Canada, believed that it is mistaken to look for justifications for patient-centered method, he believed that some concepts like "patient-centeredness" are good in themselves, and their justification lies on their moral basis (*Stewart et al., 1995*).

Several tools were developed worldwide to measure attitude toward patient centered care (*Hudon et al., 2011*; *Mead & Bower, 2000*). The Patient Practitioner Orientation Scale (PPOS) was developed by *Krupat (1996)* in the USA in 1996 and it is one of the most frequently used tools recently (*Bejarano et al., 2022*). The PPOS is used and validated in several studies from different countries (*Pauli & Wilhelmy, 2021*; *Jiang et al., 2022*; *Pereira et al., 2013*; *Hur et al., 2014*).

Medical curricula are actively embracing more training in communication skills (including patient-centeredness) (*Haidet et al., 2001*; *Haidet et al., 2002*). The Kingdom of Saudi Arabia (KSA) adopted an expansion plan for higher education, including medical education, several universities were established in the recent decades and many of them adopted curricula from developed countries and modified them to be compatible with the local needs and cultures. The Faculty of Medicine at Taibah University adopted a new curriculum from Manchester University in which learning communication skills is conducted mainly in year two and three through the Early Clinical Experience (ECE) course. In this study we aimed to examine whether this communication skills course (ECE) might support shaping medical students' attitudes towards patient-centered care using the PPOS instrument. Results from this study will serve as well in comparing Taibah students' attitudes towards patient-centeredness with other health professional students worldwide in studies used the same tool and to explore cross gender variations recognized in this context (*Madhan, Rajpurohit & Gayathri, 2011*; *Krupat et al., 1999*).

## MATERIALS AND METHODS

### Study design, setting, and data collection

This is a prospective observational study with a pre-post design, conducted at the Faculty of Medicine, Taibah University in Medina, Saudi Arabia. The study targeted a batch of medical students at the start of their second study year (before exposure to the ECE communication skills course) and the same batch was targeted again at the start of their fourth study year (after exposure to the ECE communication skills course). The original English version of the PPOS questionnaire was distributed to students on a paper form during classes. The participation in the study was totally voluntary. Clear instructions were included in the questionnaire form on the purpose of the research and how to fill in the form. Students who responded by answering less than 15 questions were excluded (following the guidelines provided by the questionnaire developers). Accordingly, two students were excluded in year two and seven were excluded in year four.

### Data collection tool

The PPOS items were crafted to assess participants' attitude towards patient-centered care. The questionnaire contains 18 items, divided into two subscales, each of nine items. The first subscale is the "sharing subscale" in which the items assess participants' attitude towards sharing information, power, and control with the patient during the clinical encounter. The second subscale is the "caring subscale", which assesses the attitude towards caring about patients' emotions, feelings, concerns, and preferences. The PPOS employs a six-point Likert scale, ranging from 6 (strongly agree) to 1 (strongly disagree). The higher the mean overall scores the more patient-centered is the respondent, while the lower overall mean scores the more physician-centered attitude. The internal consistency of the original tool has been shown to be satisfactory for the overall scores (Cronbach's $\alpha = 0.73$) (*Krupat et al., 2000*).

### Curriculum description

The current curriculum of the Faculty of Medicine at Taibah University is designed for the Medical Bachelor and Bachelor of Surgery (MBBS) program. This curriculum was originally designed at the School of Medical Sciences, University of Manchester in UK. As a part of a collaboration project, the Faculty of Medicine at the University of Taibah gained the right to utilize the curriculum. It was revised and modified to suit the local context. It is a six-year curriculum. Year 2 and year 3 are within the pre-clinical phase where problem-based learning (PBL) is the cornerstone for learning. In the clinical phase (Year 4, 5, and 6) teaching methods include mainly case-based collaborative learning and clerkships.

The Early Clinical Experience (ECE) is a longitudinal track through years two and three. It contains a series of integrated theoretical and practical simulated sessions in communication skills. Sessions start with a twenty-minute theoretical background like "listening to the patient" in communication skills, followed by 90 min practical application. Students are distributed in groups of nine guided by tutors to practice simulation. Primary care visits and hospital visits are also arranged to practice the learned

communication and clinical skills. One of the major intended learning outcomes of the ECE course is *"To be able to conduct a medical interview employing a patient-centered approach"*. Other intended outcomes of the course are to be able to conduct a basic system examination, practice certain clinical skills, perform basic concepts of cardiopulmonary resuscitation, and practice prescribing skills.

## Data analysis

Statistical analysis was conducted using the Statistical Package for Social Sciences (SPSS) version 28. The level of statistical significance was set at $p$-value of <0.05. Continuous variables were presented as means and standard deviations (SD). Independent sample T test was used to compare groups of students' responses to the PPOS. Before-and-after data was compared using collective cohort data. Effect size was measured using Cohen's d. The conventional effect sizes proposed by Cohen are as follows: 0.20 (as small effect), 0.50 (as moderate effect), and 0.8 (as large effect) (*Cohen, 2013*).

## Ethical approval

The study was approved by the Research Ethics Committee at Taibah University (Study ID: TU-20-024). Approval date: 02.05.2021. The questionnaire was anonymous and did not include the names of participants or numbers that can indicate their names. All data was kept confidential. Students were informed about the purpose of the questionnaire and its content and that their participation is voluntary. Written informed consent was obtained from the students.

## RESULTS

A total of 153 students were targeted at the beginning of their study year two, 143 of them responded to the questionnaire (response rate = 93.5%). The same group of students were targeted again when they reached their study year four. One hundred twenty-eight of them responded to the questionnaire (response rate = 89.5%). After excluding students who responded by answering less than 15 questions (according to the questionnaire guidelines), the actual number included in the study was 141 students (72 males and 69 females) in year two and 121 (65 males and 56 females) in year four.

Table 1 shows the students' progress between year two and year four in orientation towards patient-centered care measured by the Patient Practitioner Orientation Scale (PPOS) for the whole cohort. The sharing subscale showed significant positive progress from a mean score of 3.23 ± 0.53 in year two to 3.40 ± 0.60 in year four ($p = 0.015$). In contrast the caring subscale showed statistically significant decrease from 4.06 ± 0.53 to 3.76 ± 0.68 ($p < 0.001$). This was further reflected in a statistically insignificant overall decrease from 3.64 ± 0.44 to 3.58 ± 0.55 ($p = 0.282$).

When comparing the scores of male students to female students at year 2 and at year 4, the results showed no statistically significant difference in neither the caring nor the sharing component of the PPOS both in year 2 and year 4 ($p > 0.05$) (Table 2).

Table 3 shows how female students progressed in the sharing subscale between year two and year four (from 3.22 ± 0.56 to 3.46 ± 0.62) with a statistically significant difference

**Table 1 Change of mean PPOS scores before and after the ECE program for the whole cohort.**

| Dimensions | Year 2 (n = 141) Mean ± SD | Year 4 (n = 121) Mean ± SD | t | p-value | Cohen's d |
|---|---|---|---|---|---|
| Sharing | 3.23 ± 0.53 | 3.40 ± 0.60 | 2.449 | 0.015* | 0.303 |
| Caring | 4.06 ± 0.53 | 3.76 ± 0.68 | 4.041 | <0.001* | 0.501 |
| Overall | 3.64 ± 0.44 | 3.58 ± 0.55 | 1.078 | 0.282 | 0.134 |

Note:
* Statistically significant.

**Table 2 Comparison of PPOS scores for the male and female participants at year 2 and at year 4.**

| Dimensions | Year 2 (n = 141) | | | | Year 4 (n = 121) | | | |
|---|---|---|---|---|---|---|---|---|
| | Female (n = 69) (Mean ± SD) | Male (n = 72) (Mean ± SD) | t | p-value | Female (n = 56) (Mean ± SD) | Male (n = 65) (Mean ± SD) | t | p-value |
| Sharing | 3.22 ± 0.56 | 3.24 ± 0.51 | 0.226 | 0.822 | 3.46 ± 0.62 | 3.34 ± 0.59 | 1.066 | 0.289 |
| Caring | 4.01 ± 0.59 | 4.11 ± 0.48 | 1.057 | 0.292 | 3.80 ± 0.69 | 3.72 ± 0.68 | 0.585 | 0.560 |
| Overall | 3.61 ± 0.49 | 3.67 ± 0.38 | 0.782 | 0.436 | 3.63 ± 0.58 | 3.53 ± 0.52 | 0.942 | 0.348 |

**Table 3 Comparison of improvement of student PPOS scores before and after the ECE program sorted by gender.**

| Dimensions | Female | | | | Male | | | |
|---|---|---|---|---|---|---|---|---|
| | Year 2 (Mean ± SD) | Year 4 (Mean ± SD) | p-value | Cohen's d | Year 2 (Mean ± SD) | Year 4 (Mean ± SD) | p-value | Cohen's d |
| Sharing | 3.22 ± 0.56 | 3.46 ± 0.62 | 0.022* | 0.417 | 3.24 ± 0.51 | 3.34 ± 0.59 | 0.252 | 0.197 |
| Caring | 4.01 ± 0.59 | 3.80 ± 0.69 | 0.061 | 0.340 | 4.11 ± 0.48 | 3.72 ± 0.68 | <0.001* | 0.662 |
| Overall | 3.61 ± 0.49 | 3.63 ± 0.58 | 0.881 | 0.027 | 3.67 ± 0.38 | 3.53 ± 0.52 | 0.077 | 0.305 |

Note:
* Statistically significant.

($p$ = 0.022). However, there was statistically insignificant decrease in the caring subscale (from 4.01 ± 0.59 to 3.80 ± 0.69) and statistically insignificant progress in the overall scores by females (from 3.61 ± 0.49 to 3.63 ± 0.58).

For male students, a mild positive, statistically insignificant progress ($p$ = 0.252) is recognized in the sharing subscale (from 3.24 ± 0.51 to 3.34 ± 0.59). However, there was a statistically significant decrease in the caring subscale (from 4.11 ± 0.48 to 3.72 ± 0.68; $p < 0.001$) and statistically insignificant decrease in the overall scores (from 3.67 ± 0.38 to 3.53 ± 0.52; $p$ = 0.077). That means that the overall significant progress in the sharing component between year 2 and year 4 was mainly performed by female students.

## DISCUSSION

This prospective study aimed to examine whether this communication skills course (ECE) might support shaping medical students' attitudes towards patient-centered care at Taibah University, KSA. The study targeted a batch of medical students at the start of their second study year (before exposure to the ECE communication skills course) and the same batch

was targeted again at the start of their fourth study year (after exposure to the ECE communication skills course).

The results showed statistically significant progress in the sharing subscale for all participants. On the other hand, there was a statistically significant decline in the caring subscale and consequently, a decline in the overall scores of the PPOS which was statistically insignificant. This statistically significant difference in the decline of the caring scale could possibly be attributed to a negative impact of poor caring attitudes of the clinical trainers due to time and productivity pressures and lack of faculty development.

Taibah medical students' orientation towards patient-centered care is generally average when compared with other medical students in multiple published studies from other parts of the world. The students' overall scores of PPOS in this study are higher than students' scores in studies that were conducted in China (*Song et al., 2022*), Poland (*Pers et al., 2019*), and Mali (*Hurley et al., 2018*). However, PPOS scores of our students are lower in comparison to studies conducted in Brazil (*Ribeiro, Krupat & Amaral, 2007*), Korea (*Hur, Cho & Choi, 2017*), and Germany (*Pauli & Wilhelmy, 2021*). The current study showed higher scores in the caring subscale compared with the sharing sub-scale in general, which means that participants have more positive attitude towards caring about patients' emotions, feelings, concerns, and preferences, and have a less positive attitude towards sharing information, power, and control with patients. This is in favor of the paternalistic biomedical communication model. The observed trend of having higher scores in the caring subscale when compared with the sharing subscale is dominating in most previous studies (*Hurley et al., 2018*; *Ribeiro, Krupat & Amaral, 2007*; *Hur, Cho & Choi, 2017*); exceptionally, in studies from Sudan (*Mohamed et al., 2019*), and Portugal (*Manchaiah et al., 2014*) where the sharing subscale was higher. Results from this study are very close to that obtained from a study including South African students, both in the total and in the sharing and caring subscales (*Ismail et al., 2022*). According to the ASE socio-psychological model (*de Vries, Dijkstra & Kuhlman, 1988*), this disparity in patient-centered attitude scores in different settings can be attributed to many factors, including attitude, social background, and self-efficiency (ASE). Additionally, the presence of certain barriers (*Ting et al., 2016*) and learned skills (*van den Eertwegh et al., 2013*) might also affect.

Previous studies showed generally higher scores by females when compared with their male peers (*Wahlqvist et al., 2010*; *Guan et al., 2023*; *Liu et al., 2019*). In the current study, males scored higher than females in year two in all domains, while females scored higher than males in year four in all domains. However, these differences were statistically insignificant. The males' relative higher starting PPOS scores allowed less space for further progress in year four.

The significant progress in the sharing component when compared to the caring component was recognized also in a recent study from Italy (*Ardenghi et al., 2024*). Such a finding draws attention to the importance of strengthening the caring component training in the future. The study also showed a significant decline in the caring component. This decline while students advance in medical schools was recognized in previous studies

(*Haidet et al., 2002*). That was mainly attributed to exposure to clinical clerkships where the biomedical approach is more dominant with the progress in medical school.

The strength of this research work emerges from its multi-dimensional impact. It highlights communication skills as an area for research, targets curriculum development, and allows comparison of our students' orientation towards patient-centered care with students worldwide in other schools. There is scarce research on patient centeredness in our Eastern Mediterranean Region, so this study fills this research gap and provides scientific evidence-based measures in this important area. The use of the PPOS tool adds another strength to the current study since it is a validated tool and used extensively in several settings.

The study demonstrates commendable internal validity, as it employs a validated tool to track the advancement of patient-centeredness and, more importantly, utilizes an appropriate prospective design. However, caution must be exercised when generalizing our findings to other contexts, due to potential background and cultural differences. The study has also its limitations, we used the original English language version of the PPOS, assuming that there won't be a language barrier since the study language for our participants is English, it might have been more appropriate to use a translated Arabic version, which is unfortunately unavailable according to our knowledge. A pertinent question is the correlation between the students' evaluation using performance-based tools like Objective Structured Clinical Examinations/OSCE and written exams or questionnaire tools like the PPOS. This issue was discussed by *Kiessling et al. (2023)* in a systematic review on the concurrent and predictive validity of the theoretical tools. They concluded that theoretical assessment tools might help in predicting performance-based assessments to a limited extent, however, they cannot replace them totally. However, it is logistically difficult to use OSCE evaluation before and after the course for the whole batch for research purposes. Moreover, time bias was not addressed in our study, where changes in students' attitudes may have occurred over time due to factors unrelated to the course, such as maturation, increased clinical exposure, or external experiences.

This work conveys a clear message that future curriculum reforms should incorporate more opportunities for training students in the emotional (caring) aspects of patient-centered care. Alternative teaching methods should be tried for better effectiveness to foster empathy among students such as (mentorship, narrative medicine, and using professional simulated patients). Future research could include a follow-up study on graduates' "patient-centeredness" in real-life clinical practice to assess its sustainability. Additionally, to ensure that patient centeredness is properly evaluated in similar programs, triangulation of evidence using multiple tools such as performance in clinical exams and using qualitative interviews might strengthen the findings that were described in the literature. Further analysis may be needed to explore the educational and cultural factors contributing to gender differences.

## CONCLUSIONS

Taibah medical students' PPOS scores were in egalitarian relationship with studies in other countries. The communication skills course might have had a positive impact on students'

attitudes towards a patient centered approach. The attitude towards sharing information in decision-making showed statistically significant improvement after the course, but notably, the medical students' caring attitudes declined. This indicates a need for further emphasis on patient-centered communication, especially on the "caring" aspects, in the medical curriculum.

Female students showed more progress compared with males which could be attributed to the cultural differences in their roles in the community. However, the reasons for gender variation might need further in-depth analysis in future qualitative studies considering the possible cultural or educational factors that might be responsible for the observed difference.

## ACKNOWLEDGEMENTS

We want to express our gratitude to Dr. Reem Ibrahim Qabshawy for her support in data collection and to Dr. Ibrahim Alhujaili and Dr. Mohammed Almutairi for their support in data entry.

### Funding

The authors received no funding for this work.

### Competing Interests

The authors declare that there are no competing interests.

### Author Contributions

- Khalid Gaffer Mohamed conceived and designed the experiments, performed the experiments, analyzed the data, prepared figures and/or tables, authored or reviewed drafts of the article, and approved the final draft.
- Amer Almarabheh conceived and designed the experiments, analyzed the data, prepared figures and/or tables, authored or reviewed drafts of the article, and approved the final draft.
- Asim Mukhideer Almughamsi conceived and designed the experiments, performed the experiments, authored or reviewed drafts of the article, and approved the final draft.
- Hany Atwa conceived and designed the experiments, analyzed the data, prepared figures and/or tables, authored or reviewed drafts of the article, and approved the final draft.
- Mohamed Hany Shehata conceived and designed the experiments, analyzed the data, prepared figures and/or tables, authored or reviewed drafts of the article, and approved the final draft.

### Human Ethics

The following information was supplied relating to ethical approvals (*i.e.*, approving body and any reference numbers):

Ethical approval

The study is approved by the Research Ethics Committee at Taibah University (Study ID: TU-20-024). Approval date: 02.05.2021.

## Data Availability

The raw data is available in the Supplemental File.

## Supplemental Information

Supplemental information for this article can be found online at http://dx.doi.org/10.7717/peerj.18676#supplemental-information.

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
