# Peer review of "Evaluation of the effect of a communication skills course on medical students' attitude towards patient-centered care: a prospective study"

_PeerJ, doi:10.7717/peerj.18676_

## Round 0.1 · original submission · Major Revisions

I read the paper myself, and I am sorry to say that it needs to be improved.
Please follow what Reviewer 2 indicated.

·

Basic reporting

.

Experimental design

.

Validity of the findings

.

Additional comments

The review is attached.

·

Basic reporting

1. The abstract and the manuscript must be aligned.
2. The introduction is sufficient according to the history. I would like to read some comments on the local position compared with other countries' development. Professor Ian R McWhinney's belief is abruptly placed in the manuscript. Can it be moved to the discussion section and discussed? A method section in the introductions seems, as well, more appropriate put in the method section.
3. The manuscript is written in readable and acceptable English, but some corrections can improve the text, such as deleting repetitions (lines 75 to 78: .. used in many countries).

Experimental design

Focusing on medical students’ development in their journey to becoming physicians and evaluating the courses offered in the schools are relevant. It is also appropriate to use evaluation tools that produce comparable results.
1. The precise aim of the research is not clear to me. It will fill a gap in the East Mediterranean Region research on patient-centered attitude (Introduction), which was filled with “... assessment of student attitudes before and after the course using a validated tool called …” (the abstract). In the discussion, it is stated that “This prospective study aimed to evaluate the change in students’ attitude towards patient-centeredness…” A precise and consistent aim is lacking.
2. Is a single score of personal traits comparable with other countries’ measurements? As what we measure can vary by person (culture, childhood, special neurons, and maybe blood flow in some parts of the brain, etc.), we need two upon each other following measures and work with the change. These authors did this, but what did they use to compare with other countries?
3. It seems that before-and-after data was compared using collective cohort data. I think this should be stated. It can be questioned whether students’ development in different areas is normally distributed, but the authors work with standard deviation. A comment and data on this should be included to verify the method used.
4. The authors operate with statistical significance. The manuscript lacks a discussion on the relevant difference.

Validity of the findings

1. The all-over shadowing challenge for results is the time bias introduced by the method used, which must be discussed. Further, the authors need other methods to claim found causal processes.
2. The data collection tool is well described and supported by files. However, have the authors done something to meet the time bias?
3. As a precise and consistent aim is lacking, it is also unclear whether the conclusion follows the aim.

Additional comments

I would so much like to se more evaluations of medical training and education, but I am sorry to say that this manuscript still needs some work before publication. Using one or more reporting guidelines, for instance, from the Equator network home page would be a good idea.

---

## Round 0.2 · Major Revisions

Dear authors,

I am not used to allowing a second round of major revision, but I decided to give you the opportunity to align with what the reviewers asked.

Thus, please follow thoroughly what they indicate, especially addressing the input of Reviewer 2.

In the case you decide not to address what they indicated, I would be forced to reject the paper.

Best,

·

Basic reporting

Basic Reporting
The study titled "Evaluation of the effect of a communication skills course on medical students' attitude towards patient-centered care" examines the impact of targeted communication training on medical students' attitudes. Conducted at Taibah University in Saudi Arabia, this prospective observational study provides valuable insights into how structured courses influence the patient-practitioner dynamic, particularly within patient-centered care. The Patient Practitioner Orientation Scale (PPOS) was used pre and post course to evaluate changes in attitudes.

Experimental design

This prospective observational study used a repeated-measures approach, assessing the same cohort of medical students at two points in their education: the beginning of year two and the beginning of year four. This longitudinal perspective provides insights into the course's long-term impact. The PPOS scale, with two subscales: sharing (emphasizing managerial aspects of information sharing, power, and control) and caring (focusing on empathy and patient preferences) was used to measure students' attitudes toward patient-centered care. Statistical analysis involved calculating mean scores for each subscale (out of 6) and comparing them using an independent-sample T-test across the two time points. Response rates were high: 93.5% in year two and 89.5% in year four.

Validity of the findings

The study found a significant improvement in the sharing subscale but a decline in the caring subscale, indicating that students appreciated the importance of shared decision-making and collaboration post-training. However, the decline in caring, which relates to empathy and patient understanding, raises questions about the course's influence on emotional engagement. This finding suggests that while students recognize the value of information exchange and patient autonomy, additional focus is needed to nurture empathetic engagement. The authors discuss potential curriculum reforms to include tools fostering empathy and patient-centered care. The study used the validated PPOS tool to measure attitudes, ensuring measurement accuracy. By distinguishing between sharing and caring, the study offers a nuanced understanding of attitudinal changes. To reduce the risk of socially desirable responses, the researchers used reverse-worded items in the questionnaire.
Internal Validity
The prospective observational design, with repeated measures on the same cohort at two time points, strengthens the study's internal validity.
External Validity
This study was conducted at a single institution, Taibah University in Saudi Arabia, which may limit generalizability to other medical schools or cultural settings, as the local environment might uniquely influence patient-centered care attitudes. However, the high response rates across both assessment points enhance the sample's representativeness within this context. Statistically significant changes were noted in the sharing subscale, alongside a significant decline in caring. The overall scores showed no significant improvement (p = 0.282), suggesting a nuanced course impact. The authors recommend further research, including follow-up studies assessing the sustainability of sharing and caring skills among graduates in clinical practice. The authors acknowledge potential limitations and the risks of generalizing findings, also comparing results with other institutions and considering the influence of different care models (e.g., partnership vs. paternalistic).

Additional comments

The study reveals gender differences, with female students showing more progress than male students. This finding, while insightful, warrants further exploration of educational and cultural factors influencing these differences. Additionally, alternative teaching methodologies may be more effective for fostering empathy.
The study contributes significantly to discussions on enhancing patient-centered approaches in medical practice. However, relying solely on the PPOS may not capture all aspects of patient-centered care. Future research could include a follow-up study of graduate patient-centeredness in clinical settings to assess its sustainability, using triangulated evidence from tools like clinical exam performance and qualitative interviews to reinforce findings. Further analysis could examine cultural and educational factors related to gender differences.

·

Basic reporting

The paper still needs language improvement, for instance “Wholistic” = holistic

Experimental design

This paper now include the important time-bias and have improved considerable. It present interesting results. However, to be recommended for publication, some not fully addressed comment of mine must be considered:
The aim, method and conclusion must be aligned.
The methods used are not appropriate to measure effectiveness. As the authors cannot change the method used, they have to adjust the aim and the conclusion to fit the method.
The aim must be adjusted to for instance:
“In this study we aimed to examine whether this communication skills course (ECE) might support shaping medical students' attitudes towards patient-centered care.”
The conclusion must be adjusted accordingly, to for instance: “The communication skills course might have had a positive impact on students' attitudes towards a patient centered approach. The attitude towards sharing information and decision-making improved in the years after the course, but notably, the medical students’ caring attitudes declined. This indicates a need for further emphasis on patient-centered communication, especially on the caring aspects, in the medical curriculum."

Validity of the findings

No comments

Additional comments

No comments

---

## Round 0.3 · accepted · Accept

The reviewers have informed me that your work is worthy of publication on PeerJ.

·

Basic reporting

no comments

Experimental design

no comments

Validity of the findings

no comments

Additional comments

no comments